# Whole-genome sequence analysis of high-level penicillin-resistant strains and antimicrobial susceptibility of *Neisseria gonorrhoeae* clinical isolates from Thailand

**Natakorn Nokchan**[1], **Thidathip Wongsurawat**[2,3], **Piroon Jenjaroenpun**[2,3], **Perapon Nitayanon**[1], **Chanwit Tribuddharat**[1] *

**1** Department of Microbiology, Faculty of Medicine Siriraj Hospital, Mahidol University, Bangkok, Thailand, **2** Department of Biomedical Informatics, University of Arkansas for Medical Sciences, Little Rock, Arkansas, United States of America, **3** Division of Bioinformatics and Data Management for Research, Department of Research and Development, Faculty of Medicine Siriraj Hospital, Mahidol University, Bangkok, Thailand

* chanwit.tri@mahidol.ac.th

## Abstract

### Background

The increasing rate of antimicrobial-resistant *Neisseria gonorrhoeae* poses a considerable public health threat due to the difficulty in treating gonococcal infections. This study examined antimicrobial resistance (AMR) to drugs recommended for gonorrhea treatment between 2015 and 2017, and the AMR determinants and genetic compositions of plasmids in 3 gonococcal strains with high-level penicillin resistance.

### Methods

We collected 117 *N. gonorrhoeae* isolates from patients with gonococcal infections who attended Siriraj Hospital, Bangkok, Thailand, between 2015 and 2017. Minimum inhibitory concentrations (MICs) of penicillin, tetracycline, ciprofloxacin, azithromycin, spectinomycin, cefixime, and ceftriaxone were determined by the agar dilution method. PCR amplification and sequencing of 23S rRNA and *mtrR* (a negative regulator of MtrCDE efflux pump) were performed. Whole genomes of 3 PPNG strains with high-level penicillin resistance (MIC ≥ 128 μg/ml) were sequenced using Illumina and Nanopore sequencing platforms.

### Results

The proportions of *N. gonorrhoeae* isolates with resistance were 84.6% for penicillin, 91.5% for tetracycline, and 96.6% for ciprofloxacin. All isolates were susceptible to spectinomycin, azithromycin, cefixime, and ceftriaxone. An adenine deletion within a 13 bp inverted repeat sequence in the *mtrR* promoter and an H105Y mutation in the *mtrR* coding region were found in the *N. gonorrhoeae* isolate with the highest azithromycin MIC value (1 μg/ml). Three high-level penicillin-resistant isolates contained nonmosaic type II *penA* and had mutations in *penB* and the *mtrR* coding region. All isolates with high-level penicillin

**Data Availability Statement:** All relevant data are within the paper and its Supporting Information files.

**Funding:** This work was financially supported by the Thailand Research Fund through the Royal Golden Jubilee Ph.D. Program (grant number PHD/0025/2557 awarded to N.N.). The funder had no role in study design, data collection and analysis, decision to publish, or preparation of the manuscript.

**Competing interests:** The authors have declared that no competing interests exist.

resistance carried the conjugative plasmids with or without the Dutch type *tetM* determinant, the beta-lactamase plasmid (Rio/Toronto), and the cryptic plasmid.

## Conclusions

The gonococcal population in Thailand showed high susceptibility to ceftriaxone and azithromycin, current dual therapy recommended for gonorrhea treatment. As elevated MIC of azithromycin has been observed in 1 strain of *N. gonorrhoeae*, expanded and enhanced surveillance of antimicrobial susceptibility and study of genetic resistance determinants are essential to improve treatment guidelines.

## Introduction

Gonorrhea is one of the most common sexually transmitted infections globally, with an estimated 86.9 million new gonorrhea cases among adults aged 15 to 49 years annually, according to the World Health Organization (WHO) [1]. The causative agent, *Neisseria gonorrhoeae*, has become a serious public health problem and is of increasing concern due to its distinct capacity to progressively develop resistance to all known classes of antibiotics recommended for treatment, including sulfonamide, beta-lactams (penicillin and narrow-spectrum cephalosporins), tetracycline, fluoroquinolones, and macrolides [2]. *N. gonorrhoeae* was therefore classified as an urgent threat by the Centers for Disease Control and Prevention (CDC). The WHO also included it in the global priority pathogen list due to the urgent need for new antibiotics [3]. In the absence of an effective vaccine, patient management is primarily based on preventive measures and effective antibiotic treatment. Currently, the WHO recommends dual antimicrobial therapy (ceftriaxone plus azithromycin) as a first-line empirical treatment for uncomplicated gonorrhea [4]. Nonetheless, it was of great concern when gonococcal strains with reduced susceptibility or resistance to ceftriaxone or azithromycin began to be reported in multiple countries around the world [5, 6], resulting in treatment failures [7]. Gonorrhea may soon become untreatable. In 2016, the first failure of gonorrhea treatment with recommended dual therapy was recorded [8], and in 2018, gonococcal strains exhibiting ceftriaxone resistance and high-level resistance to azithromycin were identified in England and Australia [9]. The continued development of resistance in *N. gonorrhoeae* emphasizes the urgent need for improved antimicrobial resistance surveillance and new antimicrobials for gonorrhea treatment.

Penicillin, the first effective antibiotic, was officially approved for the treatment of gonococcal urethritis in 1943. Penicillin resistance in *N. gonorrhoeae* is acquired from stepwise mutations of several determinants of penicillin resistance. These are mutations on chromosomes (chromosomal-mediated resistance) and horizontal gene transfer of the beta-lactamase plasmids. The high-level penicillin resistance is conferred by plasmids, namely, Asian, African, and Rio/Toronto plasmids. The chromosomal mutations in associated penicillin resistance genes are *penA* (encoding the main lethal target penicillin-binding protein 2 [PBP2]); *ponA* (encoding PBP1); *mtrR* (encoding the transcriptional repressor of the *mtrCDE* efflux pump); *penB* (encoding a major outer membrane protein); and *pilQ* (encoding pore-forming secretin PilQ). These mutations were associated with the following AMR mechanisms:

1. Reducing the target affinity for the drug (*penA* and *ponA* mutations)

2. Increasing the efflux of the drug (*mtrR* mutation)

3. Reducing the influx of the drug (*penB* and *pilQ* mutations).

The beta-lactamase plasmids contain the $bla_{TEM-1}$ gene, which encodes a beta-lactamase capable of hydrolyzing the cyclic amide bond of penicillin's beta-lactam ring [2]. Several variants of $bla_{TEM}$ have been reported in *N. gonorrhoeae*: $bla_{TEM-135}$ (M182T substitution), $bla_{TEM-220}$ (M182T and A185T substitutions), and $bla_{TEM-239}$ [10, 11]. Only 1 additional amino acid change in $bla_{TEM-135}$, such as $bla_{TEM-20}$ (G238S), results in extended-spectrum beta-lactamase (ESBL) evolution, which can destroy the activity of third-generation cephalosporins available for gonorrhea treatment during the study period, including cefixime and ceftriaxone. For extended-spectrum cephalosporins (ESCs), the main mechanisms of reduced susceptibility or resistance were *penA* mutations (A501V and A501T) and mosaic *penA* alleles (associated with up to 70 amino acid substitutions relative to wild-type), which are acquired from interspecies recombination between *penA* genes of *N. gonorrhoeae* and commensal *Neisseria* species (*N. cinerea* and *N. perflava*) [12]. The elevated minimum inhibitory concentrations (MICs) of azithromycin result from mutations in the 23S rRNA gene (A2058G, A2059G, and C2611T), which contribute to intermediate-level or high-level resistance and, to a lesser extent, in the *mtr* (multiple transferable resistance) efflux pump locus, causing increased drug efflux [2].

Typically, *N. gonorrhoeae* carries up to 3 types of plasmids: conjugative, beta-lactamase, and cryptic. The first 2 types contribute to high-level resistance to tetracycline and penicillin, respectively. The 3 types of conjugative plasmids have been described based on *tetM* resistance determinant and restriction endonuclease mapping, including American, Dutch, and marker-less (lacking *tetM*) types. Furthermore, the difference in nucleotide sequences of the *tetM* determinants carried by American- and Dutch-type plasmids was identified, resulting in the designations "American" and "Dutch." The conjugative plasmid enables small non-self-transmissible beta-lactamase plasmids to mobilize into other bacteria, such as *Escherichia coli*. The small cryptic plasmid was frequently found in *N. gonorrhoeae*; nevertheless, its role remains unknown [13].

In Thailand, an overall gonorrhea incidence rate of 15.13 per 100 000 population was reported in 2018. However, this incidence was probably underestimated because of suboptimal diagnosis and surveillance and inadequate screening in asymptomatic patients, especially among men who have sex with men [14]. The AMR surveillance was based on the Enhanced Gonococcal Antimicrobial Surveillance Program of the WHO and the Centers for Disease Control and Prevention of the United States [15]. Siriraj Hospital, Bangkok, managed the AMR surveillance within Thailand [16]. Located in the central part of Thailand, the hospital is the largest tertiary-care university hospital in the nation. It services large numbers of patients, many of whom are referred from other hospitals. According to Tribuddharat et al. (2017), all *N. gonorrhoeae* isolates from Siriraj Hospital and Bangrak Hospital (National Center for Sexually Transmitted Infections) showed susceptibility to spectinomycin, azithromycin, and ceftriaxone, indicating that the recommended treatment of dual antimicrobial therapy for gonorrhea remained effective in Thailand [16]. No ceftriaxone-resistant gonococcal strains have been reported in Thailand to date. However, decreased susceptibility to ceftriaxone has been identified in a few gonococcal isolates, as described elsewhere [17].

Integrons are mobile genetic elements that play a pivotal role in conferring AMR and disseminating AMR determinants, particularly in Gram-negative bacterial pathogens. The fundamental structure of integrons consists of a 5'-conserved segment (5'-CS) with an integrase gene (*intI*) and an *attI* site. There is also a 3'-conserved segment (3'-CS) with a quaternary ammonium compound resistance gene (*qacEΔ1*), a sulfonamide resistance gene (*sul*), and an open reading frame of unknown function (*orf5*). Of the 5 reported integron classes, class 1 integrons are very common in isolates from clinical settings. They can capture and express

exogenous genes (antibiotic resistance, metabolic, or virulence) embedded as gene cassettes of variable regions between the 2 conserved segments [18]. According to the literature, class 1 integrons have never been examined in clinical isolates of *N. gonorrhoeae*.

With advances in technology, whole-genome sequencing has become an ideal tool in well-resourced settings as it provides more detailed information and greater accuracy than other methods [19]. It is used to (1) study the molecular epidemiology, genetic relationships, and transmission dynamics of AMR gonococcal strains; (2) predict antimicrobial susceptibilities; and (3) elucidate novel or known AMR determinants.

Relatively limited antimicrobial susceptibility data are available on *N. gonorrhoeae* in Thailand. In addition, comprehensive characterizations of beta-lactam resistance genes and mutations have not been described. More data on antimicrobial resistance in *N. gonorrhoeae* clinical isolates from Thailand are needed to inform treatment guidelines. Additional local- and national-level details on the emergence and dissemination of resistant gonococcal strains are essential for international countries to control and prevent these strains. This study had the following objectives:

1. Determine the antimicrobial susceptibility of *N. gonorrhoeae* isolates to drugs used for gonorrhea treatment in Thailand between 2015 and 2017

2. Identify point mutations that confer an elevated MIC of azithromycin in the isolate with the highest azithromycin MIC of the isolates

3. Investigate the presence of class 1 integrons

4. Uncover beta-lactam resistance determinants and analyze the plasmid composition of 3 *N. gonorrhoeae* strains with high-level penicillin resistance

## Materials and methods

### *N. gonorrhoeae* isolates and species identification

A total of 117 gonococcal isolates were obtained from Siriraj Hospital patients suspected of having gonococcal infections between January 2015 and December 2017. All isolates were collected from specimens from different anatomical body sites (the vagina, cervix, urethra, penis, fallopian tube, urine, eye, joint, and wrist) and stored in Amies medium to maintain the viability of the isolates during their transportation. *N. gonorrhoeae* culture was carried out on chocolate agar (Clinical Diagnostics Ltd., Thailand) and incubated at 35˚C ± 2˚C in a 5% $CO_2$ atmosphere for 20 to 24 hours. The identification of *N. gonorrhoeae* species was confirmed by the growth of typical colonies on chocolate agar, the presence of Gram-negative, coffee bean-shaped diplococci with polymorphonuclear leukocytes under a microscope, and the VITEK 2 *Neisseria-Haemophilus* (NH) identification card (bioMérieux Inc., Durham, NC, USA). The pure culture of *N. gonorrhoeae* isolates was preserved in brain-heart infusion broth supplemented with 30% (volume/volume) glycerol at -80˚C. The routine clinical diagnostic laboratory sourced all clinical isolates of *N. gonorrhoeae* and the *N. gonorrhoeae* ATCC 49226 used for quality control of the antimicrobial susceptibility testing.

### Beta-lactamase and antimicrobial susceptibility testing

Beta-lactamase production of *N. gonorrhoeae* isolates was tested by a nitrocefin disk (BD Diagnostics, Franklin Lakes, NJ, USA) as per the manufacturer's instructions. All beta-lactamase-positive isolates were known penicillinase-producing *N. gonorrhoeae* (PPNG) isolates.

The MICs (µg/ml) of penicillin, tetracycline, ciprofloxacin, azithromycin, spectinomycin, cefixime, and ceftriaxone were determined by the agar dilution method on GC agar base (Oxoid Ltd., Hampshire, UK) supplemented with 1% IsoVitaleX (BBL Microbiology Systems, Cockeysville, MD, USA). The antibiotic powder was purchased from Sigma–Aldrich (Milwaukee, WI, USA). The practice of conducting the agar dilution method was performed following the Clinical and Laboratory Standards Institute guidelines [20]. The agar dilution was performed in duplicate, and the obtained MIC values were interpreted based on the interpretative criteria of the CLSI guidelines [21]. *N. gonorrhoeae* ATCC 49226 was used as the quality control strain for the agar dilution test.

## Genomic DNA extraction

Total genomic DNA was extracted from *N. gonorrhoeae* isolates using the Gentra Puregene Yeast/Bacteria Kit, following the manufacturer's instructions for DNA purification from Gram-negative bacteria (Qiagen, Valencia, CA, USA). DNA concentration and purity were determined with a NanoDrop ND-100 spectrophotometer (Bio-Active Co., Ltd, Bangkok, Thailand), and DNA samples were stored at -20˚C until use.

## PCR and DNA sequencing of azithromycin resistance genes

The *N. gonorrhoeae* isolate with the highest azithromycin MIC in this study underwent PCR and DNA sequencing to identify mutations in the 23S rRNA alleles, the *mtrR* promoter (nucleotide -120 to -1), and coding regions. A two-step PCR method was used to determine mutations in the peptidyl transferase region in domain V of 4 alleles of 23S rRNA, as described previously [22]. Amplification of the *mtrR* promoter and coding regions was performed using previously described primers and PCR conditions [23]. All PCR products were purified using a NucleoSpin Gel and PCR Clean-up Kit (Macherey–Nagel Inc., Easton, PA, USA) before sending samples to Macrogen Inc. (Seoul, Korea) for DNA sequencing. DNA gene sequences were aligned with their wild-type gene sequences obtained from the pansusceptible *N. gonorrhoeae* strain FA1090 (GenBank accession number AE004969.1) using BioEdit version 7.2.5 to determine the mutations in these genes.

## Detection of class 1 integron-integrase gene

PCR was used to determine the presence of class 1 integron-integrase gene in all *N. gonorrhoeae* clinical isolates using IntI1F and IntI1R primers as described elsewhere [24].

## Whole-genome sequencing

In order to obtain the complete genome sequence, whole-genome sequencing with both Illumina (Illumina Inc., San Diego, CA, USA) and Nanopore (Oxford Nanopore Technologies Ltd. [ONT], Oxford, UK) platforms was performed. Three gonococcal strains were randomly selected based on different *N. gonorrhoeae* multiantigen sequence typing (NG-MAST) sequence types (STs). The strains also exhibited high-level penicillin resistance (MIC $\geq$ 128 µg/ml). The characteristics of the strains are summarized in Table 1. DNA libraries for Nanopore sequencing were prepared using the rapid barcoding sequencing kit (SQK-RBK004) without DNA size selection and sequenced on a MinION device (version Mk1B; ONT) using a flow cell (FLO-MIN106, v.R9.4, ONT) for 48 h. The NEBNext Ultra II DNA Library Prep Kit (New England Biolabs, Ipswich, USA) was used to generate 150-bp paired-end libraries for the Illumina sequencing, and the libraries were sequenced on a Nova-Seq 6000 sequencer (Illumina). All raw reads generated in this study were submitted to the

**Table 1. Characteristics of *N. gonorrhoeae* isolates (n = 3) with high-level resistance to penicillin (MIC ≥ 128 μg/ml).**

| Strain number | Penicillin MIC (μg/ml) | NG-MAST |
|---|---|---|
| CT530 | 256 | ST18929 |
| CT532 | 128 | STnovel10 |
| CT602 | 256 | STnovel68 |

Abbreviations: MIC, minimum inhibitory concentration; NG-MAST, *Neisseria gonorrhoeae* multiantigen sequence typing; ST, sequence type.

National Center for Biotechnology Information (NCBI) Sequence Read Archive (SRA) under BioProject number PRJNA600334.

## Sequence assembly and bioinformatic analyses

With ONT, the base-calling of raw signals and demultiplexing were conducted using Guppy v.3.2.4. Adapter trimming was performed using Porechop v.0.2.4 (https://github.com/rrwick/Porechop). Raw reads with a short length (< 1000 bases) or low quality (mean quality score < 8) were removed with NanoFilt v.2.5.0 [25]. In the case of Illumina, the raw reads were trimmed and filtered using fastp v.0.19.5 [26]. The N50 read length was calculated using the Assembly Stats tool (https://github.com/sanger-pathogens/assembly-stats). *De novo* whole-genome assembly was carried out using the hybrid assembler Unicycler v.0.4.4 [27]. Genome sequencing and assembly statistics for these 3 *N. gonorrhoeae* genomes are summarized in S1 Table. Genome error correction, circularization, and rotation (using the *dnaA* gene as the starting point) were performed using Unicycler v.0.4.4. The quality of complete genome assemblies was evaluated with QUAST v.5.0.2 [28]. The genomic GC content was ascertained with a GC Content Calculator (https://www.sciencebuddies.org/science-fair-projects/references/genomics-g-c-content-calculator). The default parameter settings of all programs were used unless otherwise noted. Genome annotations were executed with the NCBI Prokaryotic Genome Annotation Pipeline v.4.11 [29].

The plasmid maps were generated and visualized using SnapGene Viewer software (http://www.snapgene.com/). Beta-lactam resistance genes and their mutations were predicted using Resistance Gene Identifier software (v.5.1.0, https://github.com/arpcard/rgi) against the Comprehensive Antibiotic Resistance Database (v.3.0.9, https://card.mcmaster.ca/) [30] and manually confirmed by alignment with their wild-type alleles from the *N. gonorrhoeae* strain FA1090 (GenBank accession number AE004969.1). These genes were probably related to decreased susceptibility or resistance to either penicillin or ESCs. They were *ponA*, *penA*, *dacB*, *pbpG*, *dacC*, *mtrR*, *pilQ*, *penB*, and $bla_{TEM-1}$. The presence of a 57-kb gonococcal genetic island (GGI) encoding a type IV secretion system (T4SS) and the type of *tetM* determinants were examined by *in silico* PCR using SnapGene software version 6.0.5 (from GSL Biotech; available at snapgene.com) [31, 32]. The conjugal plasmid backbones were subjected to *in silico Bgl*I digestion using SnapGene software version 6.0.5 (from GSL Biotech; available at snapgene.com) to differentiate between the American and Dutch type.

Multilocus sequence typing (MLST), NG-MAST, and *N. gonorrhoeae* sequence typing for antimicrobial resistance (NG-STAR) sequence types were assigned using the Bacterial Isolate Genome Sequence Database (BIGSdb) platform of the PubMLST *Neisseria* database (https://pubmlst.org/neisseria/).

## Ethical considerations

The Siriraj Hospital Institutional Review Board approved this nonclinical research study before its commencement (certificates of approval Si479/2015 and Si720/2018). Informed consent was exempted by the Review Board because our retrospective and non-clinical study used bacterial isolates recovered from clinical samples taken as part of routine clinical laboratory examination and patient information was anonymized and deidentified prior to analysis.

## Results

### Patient data and *N. gonorrhoeae* isolates

Of the 117 patients with gonorrhea, 54.7% were women, and 45.3% were men (ratio, 1:1.2). The ages of 2 patients were unknown. The median age was 20 years, ranging from 15 days to 65 years. The highest prevalence of gonorrhea (20.9%) was found in patients aged 15 to 19. The samples were recovered from the urethra (46.1%), cervix (29.1%), vagina (17.9%), eyes (4.3%), and other sites (2.6%).

### Prevalence and penicillin susceptibility of PPNG isolates

Of the 117 gonococcal isolates, 100 (85.5%) were PPNG isolates (39 in 2015, 33 in 2016, and 28 in 2017). The remaining isolates (17/117; 14.5%) were negative for beta-lactamase production. The ratio between the PPNG and non-PPNG isolates was approximately 5.9. The prevalence of PPNG isolates was 86.7% (39/45) in 2015, 89.2% (33/37) in 2016, and 80.0% (28/35) in 2017. The antimicrobial susceptibility testing revealed that almost all PPNG isolates (99%) were resistant to penicillin except one, which showed intermediate resistance. Most non-PPNG isolates (82.4%) had intermediate susceptibility to penicillin, whereas the remaining isolates (17.6%) were susceptible to penicillin.

### Antimicrobial susceptibility testing

Most gonococcal isolates exhibited resistance to penicillin (84.6%), tetracycline (91.5%), and ciprofloxacin (96.6%), while all were susceptible to spectinomycin, cefixime, ceftriaxone, and azithromycin. None of the isolates were susceptible to tetracycline. Table 2 summarizes the antimicrobial susceptibility of the *N. gonorrhoeae* isolates. Regarding the MIC distributions, penicillin resistance (MIC of 64 μg/ml) was commonly found in *N. gonorrhoeae* isolates (23/117; 19.7%). The most common MIC values for tetracycline and ciprofloxacin were 16 μg/ml

**Table 2. Antimicrobial susceptibility of 117 *N. gonorrhoeae* clinical isolates to 7 antimicrobials previously or currently used for gonorrhea treatment.**

| Antimicrobial | No. (%) of isolates | | | MIC (μg/ml) | | |
|---|---|---|---|---|---|---|
| | Susceptible | Intermediate | Resistant | Range | $MIC_{50}$ | $MIC_{90}$ |
| Penicillin G | 3 (2.6) | 15 (12.8) | 99 (84.6) | 0.06–256 | 16 | 128 |
| Tetracycline | 0 | 10 (8.5) | 107 (91.5) | 0.5–64 | 16 | 32 |
| Ciprofloxacin | 3 (2.6) | 1 (0.8) | 113 (96.6) | $\leq 0.001$–16 | 2 | 8 |
| Azithromycin | 117 (100) | 0 | 0 | $\leq 0.004$–1 | 0.06 | 0.12 |
| Spectinomycin | 117 (100) | 0 | 0 | 2–32 | 16 | 32 |
| Cefixime | 117 (100) | 0 | 0 | $\leq 0.002$–0.06 | 0.008 | 0.016 |
| Ceftriaxone | 117 (100) | 0 | 0 | $\leq 0.002$–0.03 | 0.002 | 0.004 |

Abbreviations: MIC, minimum inhibitory concentration; $MIC_{50}$, minimum inhibitory concentration for 50% of isolates; $MIC_{90}$, minimum inhibitory concentration for 90% of isolates

(61/117; 52.1%) and 2 μg/ml (68/117; 58.1%), respectively. In the case of spectinomycin, the proportions of *N. gonorrhoeae* isolates with MIC values of 16 μg/ml and 32 μg/ml were 42.7% (50/117) and 43.6% (51/117), respectively. An azithromycin MIC of 0.06 μg/ml was frequently observed in the gonococcal isolates (45/117; 38.5%). The prevalences of *N. gonorrhoeae* isolates susceptible to cefixime (MIC = 0.008) and ceftriaxone (MIC ≤ 0.002) were 41.9% (49/117) and 78.6% (92/117), respectively. The lowest spectinomycin MIC increased every year between 2015 and 2017. The highest azithromycin MIC (1 μg/ml) was observed in *N. gonorrhoeae* obtained in 2017.

## Characterization of azithromycin resistance determinants

As a result of antimicrobial susceptibility, *N. gonorrhoeae* isolate number 577 had the highest azithromycin MIC of 1 μg/ml, representing borderline resistance to azithromycin. Therefore, this isolate was selected to examine mutations in the 23S rRNA and *mtrR* genes as these determinants are associated with azithromycin resistance. The results showed that no mutation was detected in each 23S rRNA allele (GenBank accession numbers ON638953 to ON638956). However, an A (adenine) deletion was identified in the 13-bp inverted repeat region (5′–aAAAAGTCTTTTT–3′; the deletion is indicated by lowercase letter) between the − 10 and − 35 hexamers of the *mtrR* promoter region, and an H105Y substitution was found in the *mtrR* coding region (GenBank accession number ON638952).

## Class 1 integrons, molecular epidemiology, and genetic resistance determinants

None of the *N. gonorrhoeae* clinical isolates carried class 1 integrons. The molecular epidemiology and genetic resistance determinants of 3 high-level penicillin-resistant strains were revealed by whole-genome sequencing (Table 3). *N. gonorrhoeae* strains CT530 and CT602 belonged to MLST ST8143, and the CT532 strain was assigned to MLST ST1925. Two novel and 1 known NG-MAST ST were identified: STnovel10 for CT532, STnovel68 for CT602, and ST18929 for CT530. NG-STAR analysis revealed that the *N. gonorrhoeae* strains CT530,

**Table 3. Genetic characteristics of *Neisseria gonorrhoeae* with high level resistance to penicillin (MIC ≥ 128 μg/ml).**

| Strain characteristics | CT530 | CT532 | CT602 |
|---|---|---|---|
| MLST | ST8143 | ST1925 | ST8143 |
| NG-MAST | ST18929 | STnovel10 | STnovel68 |
| NG-STAR | ST1900 | ST1663 | ST1849 |
| GGI | Absent | Absent | Absent |
| *ponA* | WT | A375T | A375T |
| *penA* | Non-mosaic type II | Non-mosaic type II | Non-mosaic type II |
| *dacB* | WT | WT | WT |
| *pbpG* | WT | WT | WT |
| *dacC* | S189A | WT | WT |
| *pilQ* | type VI | type VI | type VI |
| *porB1b* | G120K, A121N | G120K, A121N | G120R, A121D |
| *mtrR* | A39T | A39T, G162a | A39T |
| *mtrR* promoter | WT | WT | WT |
| *bla*$_{TEM}$ | M182T | M182T | M182T |

Abbreviations: GGI, gonococcal genetic island; MLST, multilocus sequence typing; NG-MAST, *Neisseria gonorrhoeae* multiantigen sequence typing; NG-STAR, *Neisseria gonorrhoeae* Sequence Typing for Antimicrobial Resistance; ST, sequence type; WT, wild type.

CT532, and CT602 were associated with NG-STAR ST1900, ST1663, and ST1849, respectively. The 57-kb GGI was not found in any of the 3 strains.

Regarding the determinants of penicillin resistance located on the chromosome, alterations in high-molecular-weight PBPs, including PBP1 encoded by *ponA* and PBP2 encoded by *penA*, were the main contributors to penicillin resistance in *N. gonorrhoeae*. Bioinformatic analyses demonstrated that the *N. gonorrhoeae* strains CT532 and CT602 contained the A375T substitution in *ponA*. In particular, there was no *ponA* mutation in the CT530 strain of *N. gonorrhoeae*. The *penA* alleles of all 3 strains were classified as nonmosaic *penA* type II due to having an insertion of aspartic acid after position 346 (D346a), F504L, A510V, and A516G alterations. The reduced extent of penicillin resistance might result from the mutations in *dacB* and *pbpG*, encoding low-molecular-weight PBPs, ie, PBP3 and PBP4, respectively; however, no mutation was found in these genes. Only the S189A mutation was found in the newly identified *dacC* gene of strain CT530, which encodes a low-molecular-weight PBP.

The promoter of *mtrR*, the gene encoding the multiple transferable resistance (Mtr) efflux pump, had no substitution in all strains. The 3 gonococcal strains showed the A39T mutation in the *mtrR* coding region, of which strain CT532 had an additional insertion of glycine after position 162 (G162a). Regarding the *pilQ* gene, which encodes the pore-forming secretin PilQ associated with type IV pilus formation, all gonococcal strains presented type VI PilQ, comprising a QAATPAKQ insertion at position 180 (180QAATPAKQ insertion), and substitutions of S341N and N648S. As to the *penB* gene (encoding pore-forming transmembrane porin), the 3 strains expressed PorB1b (but not PorB1a); PorB1b was previously confirmed to be associated with penicillin resistance [33]. Mutations in *penB* at positions 120 and 121 were detected in *N. gonorrhoeae* strains CT530 and CT532 (G120K and A121N) and in CT602 (G120R and A121D). Importantly, these strains harbored a beta-lactamase plasmid that encodes TEM-135 beta-lactamase, which is responsible for high-level penicillin resistance.

## Sequence analysis of gonococcal plasmids

Analysis of the whole-genome sequencing data revealed that each strain of high-level penicillin-resistant *N. gonorrhoeae* had 3 types of plasmids: conjugative, beta-lactamase, and cryptic. The sizes of these plasmids are listed in S1 Table. As the 3 plasmid types of each *N. gonorrhoeae* strain were very similar, the gonococcal plasmids from *N. gonorrhoeae* strain CT530 were used to represent the common features of each plasmid type (Fig 1). Nonetheless, this was not appropriate for the conjugative plasmid of strain CT602 because it did not contain the *tetM* determinant (unlike the others; Fig 1A). The conjugative plasmids of the *N. gonorrhoeae* strains carried Dutch-type *tetM* determinants and showed a similar digestion pattern to the previously reported Dutch type plasmid (GenBank accession number GU479466) after *in silico* digestion with *Bgl*I [34]. The conjugative plasmids consisted of several open reading frames (ORFs) grouped into 5 modules as follows: (1) replication initiation (*ssb* and *trfA*); (2) conjugative transfer (*tra*); (3) mating pair formation (*trb*); (4) plasmid inheritance and control (*kor*, *kle*, *inc*, and *kfr*); and (5) accessory genes (Zeta/Epsilon toxin-antitoxin system, *vapD*, and *tetM*). We identified 10 *tra* genes and 13 *trb* genes in all 3 conjugative plasmids (without *tetM* from strain CT602 but with *tetM* from strains CT530 and CT532). Two genes found in the replication initiation region, *ssb* and *trfA*, encode a single-stranded DNA-binding protein and the *oriV* activator, respectively. The plasmid inheritance and control module comprised *incC2*, which encodes the ParA ATPase responsible for the partitioning system. The *vapD* and *marR* genes located in the accessory gene region encode the virulence-associated protein D or VapD toxin and a transcriptional regulator of the MarR family, respectively. This region was important due to its *tetM* resistance determinant that encodes a cytoplasmic ribosome protecting

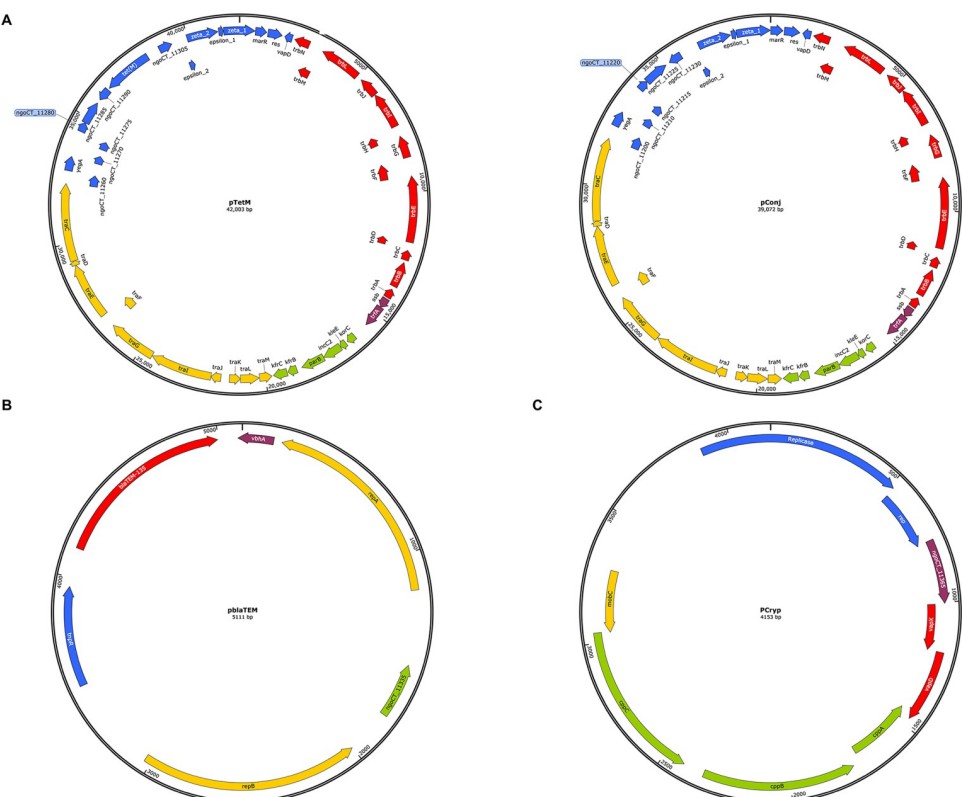

**Fig 1. Schematic representation of gonococcal plasmids.** (A) Gene organization of 2 different conjugative plasmids (upper left and right); the conjugative plasmid carrying a Dutch type *tetM* determinant and a Dutch type backbone of *N. gonorrhoeae* strain CT530 and the conjugative plasmid with no *tetM* resistance determinant of gonococcal strain CT602. The colors depict the functional modules as follows: red, mating pair formation; purple, replication initiation; green, inheritance/control; yellow, conjugative transfer; blue, accessory genes. (B) Gene organization of the beta-lactamase plasmid (Rio/Toronto type) of *N. gonorrhoeae* CT530. The colors represent the following: yellow, replication initiation; blue, recombination; red, ampicillin resistance; purple, antitoxin; green, unknown function. (C) Gene organization of the cryptic plasmid of gonococcal strain CT530. The open reading frames are colored as follows: green, cryptic plasmid proteins; yellow, plasmid mobilization; purple, proteins with unknown function; blue, plasmid-replicase genes; red, toxin-antitoxin genes.

protein that confers resistance to tetracycline. The *yegA*, found in the same region of *tetM*, encodes a DUF882 domain-containing protein. The ngoCT_11260 gene encodes a transcription elongation factor, while ngoCT_11285 encodes a site-specific DNA methyltransferase. The remaining open reading frames encode hypothetical proteins. The *tetM*-containing conjugative plasmid differed from the conjugative plasmid lacking *tetM* by the presence of the *tetM* determinant and the ngoCT_11305 gene located upstream of the start site of the *tetM* gene.

The beta-lactamase plasmids of the 3 strains were identified as the Rio/Toronto plasmid. This plasmid contained *repA* and *repB* (encoding replication initiation proteins), recombinase family protein (Tn3 resolvase), *vbhA* (encoding the antitoxin VbhA family protein), and the ngoCT_11335 gene (encoding a hypothetical protein). The $bla_{TEM}$ variant with the M182T mutation, $bla_{TEM-135}$, was identified in the Rio/Toronto plasmids of the 3 strains. This gene encodes TEM-135 beta-lactamase, which could render benzylpenicillin and ampicillin inactive.

The cryptic plasmid had 9 open reading frames; 3 of these encode cryptic plasmid proteins (*cppA*, *cppB*, and *cppC*). The *vapD* and *vapX* encode VapD toxin and VapX antitoxin, respectively. Two open reading frames encode proteins responsible for plasmid replicase, while the

*mobC* encodes the plasmid mobilization relaxosome protein. The ngoCT_11365 gene encoding a hypothetical protein was also detected in this cryptic plasmid.

## Discussion

This work described the AMR data of clinical isolates of *N. gonorrhoeae* collected from 2015 to 2017 in Thailand. We also analyzed resistance determinants and plasmids of gonococcal isolates with high levels of penicillin resistance. Additionally, the presence of class 1 integrons in these clinical isolates was examined.

Penicillin was the first-line treatment for gonorrhea in Thailand until the 1980s when the rate of PPNG isolates was 71% [35]. However, a high prevalence of PPNG isolates has since been observed. In the current investigation, 85.5% of gonococcal isolates from 2015 to 2017 were PPNG isolates; these are highly resistant to penicillin. This proportion increased from the 83.6% recorded between 2005 and 2009 [36] and from the 83.8% reported from 2008 to 2014 [16]. Intermediate resistance to penicillin of most non-PPNG isolates may be attributed to chromosome-mediated resistance by point mutations. Moreover, it was previously noted that only 1 PPNG isolate presented intermediate resistance to penicillin, whereas the remaining PPNG isolates were resistant. The possible explanations for this result may be different levels of $bla_{TEM}$ gene expression or the Canadian plasmid carrying a unique 6 bp deletion in $bla_{TEM-1}$ encoding a truncated 24 kDA TEM-1 beta-lactamase with slow activity [37].

The present work carried out agar dilution instead of the disc diffusion and E-testing used in earlier studies [16]. Agar dilution was preferred because our objective was to undertake quantitative interpretations of the MIC values of the antibiotics with as high an accuracy and reliability as possible. As expected, a high proportion of *N. gonorrhoeae* isolates were resistant to penicillin (84.6%), tetracycline (91.5%), and ciprofloxacin (96.6%). These findings are consistent with earlier data from Thailand [16]. The results suggested a very low possibility of reapplying the antibiotics previously used for gonorrhea treatment.

Interestingly, all gonococcal isolates were susceptible to spectinomycin, azithromycin, cefixime, and ceftriaxone. This finding was despite the highest cefixime and azithromycin MICs having increased only slightly after 2008–2014 (cefixime: from 0.023 to 0.06 μg/ml; azithromycin: from 0.25 to 1 μg/ml) [16]. The increase in MICs was partly due to the use of different methods for antimicrobial susceptibility testing (E-testing and agar dilution). Another factor that may have been involved was the implementation of combination ceftriaxone and azithromycin therapy, which exerts antimicrobial selective pressure, to treat gonorrhea in Thailand.

The $MIC_{50}$ and $MIC_{90}$ values of the drugs used for gonorrhea treatment (spectinomycin, azithromycin, cefixime, and ceftriaxone) were low, suggesting the high efficacy of these antibiotics against gonococcal isolates. Spectinomycin has not been available in Thailand for many years following the Thai Food and Drug Administration withdrawing its approval for use [14]. The withdrawal may explain why spectinomycin-resistant isolates were not observed in this study. At the same time, there have been no published reports of *N. gonorrhoeae* resistance to cefixime or ceftriaxone in Thailand. Consequently, our findings support the ongoing use of the recommended dual therapy of intramuscular ceftriaxone and oral azithromycin for gonorrhea treatment in Thailand. However, another possible cause of the absence of resistance to cefixime or ceftriaxone might be that local AMR surveillance is inadequate and not nationwide. It was of concern when *N. gonorrhoeae* isolates with reduced susceptibility to ceftriaxone were recently identified in Thailand by Kueakulpattana et al. [17]. In addition, decreased susceptibility or resistance to ceftriaxone and azithromycin, as a cause of treatment failures, has been sporadically described in multiple countries [6, 7].

The AMR determinants that conferred elevated MICs of azithromycin, including 23S rRNA and *mtrR*, were examined because azithromycin is part of dual therapy for the treatment of gonorrhea. Of the 117 gonococcal isolates, only one showed borderline resistance to azithromycin (an MIC of 1 μg/ml). Moreover, mutations in these resistance determinants have never been previously uncovered in Thai gonococcal isolates. The results showed a deletion of A within a 13-bp inverted-repeat sequence of the *mtrR* promoter, which can abolish the expression of *mtrR* to repress the expression of the *mtrCDE*-encoded multidrug efflux pump. This produces a higher expression of the *mtrCDE* operon, mainly due to the increased binding affinity of RNA polymerase to the *mtrCDE* promoter and a decreased susceptibility to azithromycin (to approximately a tenth) [38, 39]. In addition to an alteration in the *mtrR* promoter, H105Y substitution was identified in the *mtrR* coding region. The substitution contributes to decreased binding of MtrR to its target promoter due to the altered structure of the MtrR dimer [40]. The MIC value of azithromycin was elevated by the H105Y alteration (by approximately 2- to 4-fold) [41]. Three mutations in the peptidyl transferase loop in domain V of the 23S rRNA that influence the secondary structure of 23S rRNA (C2611T, A2058G, and A2059G) have not been identified in all 4 copies of 23S rRNA from this isolate. The C2611T mutation is associated with low- and moderate-level azithromycin resistance, whereas both A2058G and A2059G confer high-level resistance to azithromycin [22, 42].

We investigated the presence of class 1 integrons in all *N. gonorrhoeae* isolates. These bacteria display AMR to many classes of antibiotics, express natural competence, and exhibit a high frequency of DNA recombination [2]. Furthermore, class 1 integrons were previously detected in some bacteria in the genital area [43], indicating the high possibility of acquired integrons in *N. gonorrhoeae*. Surprisingly, none of our gonococcal isolates contained class 1 integrons, except the *Schaalia turicensis* isolate co-isolated with *N. gonorrhoeae* [44]. This finding supports the notion that antimicrobial resistance in *N. gonorrhoeae* is derived from chromosomal mutations of resistance genes or plasmids carrying resistance determinants. This finding is consistent with other reports and information from the NCBI genome database [10].

Whole-genome sequencing was performed to obtain data on antibiotic resistance determinants, genetic relationships, and plasmid composition of *N. gonorrhoeae* isolates with high-level resistance to penicillin as reference data for strains from Thailand. From the whole-genome sequencing analysis, *N. gonorrhoeae* strains CT530 and CT602 belonged to MLST ST8143. These initially emerged in Taiwan in 2012 and were monitored whether they would follow the pattern of MLST ST1901 (strain exhibiting higher MICs for both ceftriaxone and azithromycin) to become highly prevalent locally and globally [45]. There was no epidemiological link to strains from other countries for the remaining isolates. The 57-kb GGI, responsible for the AMR dissemination and increased fitness of *N. gonorrhoeae*, has not been found in all gonococcal strains with high-level resistance to penicillin, although it was highly prevalent in strains from other countries [46]. This genetic element was associated with the international spread of the MLST ST1901 strain that displayed decreased susceptibility to ESCs in the Western Pacific Region [46].

Concerning beta-lactam resistance, alterations in the *ponA*, *penA*, *mtrR*, *pilQ*, and *penB* genes combined with the presence of a beta-lactamase plasmid containing the *bla*$_{TEM}$ allele were identified. For *ponA* encoding PBP1, strains CT532 and CT602 harbored the A375T mutation, which was rarely found in *N. gonorrhoeae* isolates and appeared not to be associated with decreased susceptibility to ceftriaxone [47]. Although the association of this substitution with penicillin resistance has not yet been elucidated, the mutation position was close to the conserved active site motif of PBP1 (Ser-461 of the SXXK motif). Moreover, the amino acid change from alanine to threonine might affect the structure of the active site by forming hydrogen bonds with specific amino acid residues [48, 49]. The common mutation L421P,

which increases penicillin MICs slightly by 3- to 4-fold relative to wild-type, was not found in all strains [48] and was not associated with reduced susceptibility to ceftriaxone [47]. The *penA* allele is classified as nonmosaic or mosaic, followed by the type of *penA* allele assigned to Arabic numerals according to the amino acid profiles. After Arabic numerals, a decimal number is added to indicate an existing *penA* allele with different DNA sequences [50]. The *penA* alleles of all gonococcal strains were nonmosaic *penA*-2 alleles containing the D346a, F504L, A510V, and A516G substitutions. The D346a mutation was found in most penicillin-resistant strains because this mutation impeded antibiotic binding or breakage of the beta-lactam ring, leading to a decrease in the penicillin acylation rate to approximately a sixth [51]. The F504L mutation reduces the rate of penicillin acylation by restricting the conformational flexibility of PBP2 [52] and has a low impact on cephalosporin MIC [53]. Next, the A510V alteration was detected in gonococcal strains with high-level resistance to penicillin and had little effect on penicillin MIC [54]. The penicillin acylation rate was previously found to be decreased to less than a half relative to the wild-type because of the A516G mutation [54]. The nonmosaic *penA*-2 allele was the most common type and the most allelic diversity found in *N. gonorrhoeae* [50]. This *penA* allele confers resistance to penicillin but not ESC [55]. We further investigated the mutations in the *dacB* gene encoding the low-molecular-weight PBP, PBP3. PBP3 plays a role in peptidoglycan modification and recycling, and previous studies have demonstrated increased susceptibility to ceftazidime (after inactivation of DacB) and ampicillin, ceftazidime, and imipenem (after the loss of both DacB and DacC [encoded by *dacC*]) [56]. Nonetheless, we found no mutation in this gene. Likewise, there was no mutation in the *pbpG* gene encoding PBP4 in any of the *N. gonorrhoeae* strains. The *dacC* gene of *N. gonorrhoeae* strain CT530, encoding a third low molecular mass PBP that possesses carboxypeptidase and lacks conserved active site motifs, contained the S189A substitution [55]. Although the effect of this mutation is still unknown, DacC has been found to play a role in the intrinsic level of beta-lactam resistance in *P. aeruginosa* by both beta-lactam hydrolysis and trapping [57]. All strains had the A39T mutation located on the DNA binding domain of MtrR, causing the lower binding of the MtrR protein to the *mtrCDE* promoter and thus resulting in increased resistance to hydrophobic antibiotics [47]. Only the *mtrR* coding region of strain CT532 contained G162a, and its role remained unidentified. The PilQ of all *N. gonorrhoeae* strains belonged to type VI PilQ (180QAATPAKQ insertion, S341N, and N648S), and this PilQ type had no association with penicillin resistance. A previous investigation found that the E666K mutation in the *pilQ* gene was involved in penicillin resistance only when presented with alterations in *penA*, *mtrR*, and *penB* [58]. *N. gonorrhoeae* fundamentally expresses 1 of 2 forms of porin: PorB1a or PorB1b. Only the substitutions at positions 120 and 121 in PorB1b contribute to the increasing MICs of beta-lactam antibiotics and tetracyclines by replacing charged amino acids, such as aspartate (D), lysine (K), and arginine (R) [59]. The mutations found in our strains were G120K, G120R, A121D, and A121N.

Regarding plasmid-mediated resistance, $bla_{TEM-135}$, first identified in Thailand, encodes TEM-135 beta-lactamase and differs from $bla_{TEM-1}$ by 1 mutation, M182T [36]. This mutation increases enzyme thermal stability and suppresses enzyme misfolding, proteolysis, and aggregation [60, 61]. We reported the presence of beta-lactamase plasmids (Rio/Toronto plasmids) comprising the $bla_{TEM-135}$ allele in all strains with high levels of penicillin resistance. Although the driver of $bla_{TEM-135}$ evolution is still inconclusive, the $bla_{TEM-135}$ allele can evolve into TEM-type ESBLs that destroy ESCs (TEM-20 [G238S], TEM-106 [E104K], or TEM-126 [D179E]) by acquiring 1 additional mutation at those specific positions.

In summary, the high-level resistance to penicillin in *N. gonorrhoeae* strains resulted from alterations in *penA* (D346a, F504L, A510V, and A516G), *mtrR* (A39T), and *penB* resistance genes (amino acid substitutions at positions 120 and 121) as well as the presence of Rio/

Toronto plasmids carrying the $bla_{\text{TEM-135}}$ allele. It is noteworthy that the A375T substitution in the *ponA* gene was identified in our 2 *N. gonorrhoeae* strains with high-level resistance to penicillin. Consequently, further investigation of this mutation related to penicillin resistance is needed.

We provide a genomic analysis of all plasmids in *N. gonorrhoeae* strains from Thailand. The results from plasmid analysis showed that all strains carried conjugative plasmids with the Dutch type *tetM* determinant and Dutch type plasmid backbones, beta-lactamase (Rio/Toronto plasmids), and cryptic plasmids, except for strain CT602, which carried conjugative plasmids without the *tetM* determinant. The gene composition of all types of plasmids in this study was indistinguishable from previous reports [62]. The plasmid backbones of conjugative plasmids could be discriminated into Dutch type or American type due to dissimilar restriction endonuclease patterns. All 3 strains carried the beta-lactamase plasmids harboring the $bla_{\text{TEM-135}}$ allele with an amino acid substitution from methionine to threonine at position 182 (M182T), which could develop into an ESBL if it acquires 1 additional mutation. These plasmids were of the Rio/Toronto type. It is the derivative of the prototypical Asia plasmid (7.4 kb) and (as observed in our strains) confers high-level resistance to penicillin. Beta-lactamase plasmids were commonly found along with conjugative plasmids due to the capability of conjugative plasmids to comobilize beta-lactamase plasmids. Most gonococci contain the cryptic plasmid, and the absence of this plasmid has been reported to be associated with a novel chromosomal type IV secretion system found in *N. gonorrhoeae* [62].

This study has some limitations. First, the samples were only taken from one center, the routine microbiology laboratory at Siriraj Hospital, Bangkok. This results in a low number of gonococcal isolates each year and limited information on *N. gonorrhoeae* for Thailand. Additionally, since demographic, behavioral, and clinical data (including previous antibiotic exposure) of the patients were largely unavailable, it was not possible to identify the risk factors in patients that may have been associated with gonococcal infections or to correlate patient characteristics with specific patterns of resistance in *N. gonorrhoeae*.

## Conclusions

The majority of *N. gonorrhoeae* isolates in Thailand from 2015 to 2017 were resistant to previously used antibiotics for gonorrhea treatment, including penicillin, tetracycline, and ciprofloxacin. No *N. gonorrhoeae* isolates were resistant to spectinomycin, azithromycin, cefixime, and ceftriaxone, warranting the continued use of ceftriaxone plus azithromycin as the first-line treatment for gonorrhea in Thailand. Nonetheless, borderline resistance to azithromycin (MIC of 1 μg/ml) was observed in 1 gonococcal isolate, with mutations in the *mtrR* promoter (A deletion) and coding regions (H105Y). The PPNG isolates were highly prevalent (85.5%), and almost all PPNG isolates were resistant to penicillin. No class 1 integrons were identified in our gonococcal isolates. Genetic analysis of *N. gonorrhoeae* with high-level resistance to penicillin revealed that the mutations in *penA* (D346a, F504L, A510V, and A516G), *mtrR* (A39T), and *penB* determinants (G120K, G120R, A121N, and A121D) and the presence of Rio/Toronto plasmids with the $bla_{\text{TEM-135}}$ allele, contributed to a high level of penicillin resistance. The A375T mutation in *ponA* was identified as a putative mutation for penicillin resistance. Three types of plasmids were identified in *N. gonorrhoeae* isolates (conjugative, beta-lactamase, and cryptic plasmids), and they had the same gene composition as reported elsewhere [62]. Our results indicate that sustained and expanded AMR surveillance coupled with the characterization of AMR resistance determinants are critical. Doing so will enable the early detection of emergent isolates resistant to ceftriaxone or azithromycin, inform treatment guidelines, and enhance the understanding of AMR evolution.

## Supporting information

**S1 Table. Genome sequencing and assembly statistics for 3 *Neisseria gonorrhoeae* strains with high-level resistance to penicillin.**
(DOCX)

**S2 Table. Specimen source and antimicrobial susceptibility of *Neisseria gonorrhoeae* isolates from Thailand, 2015–2017.**
(XLSX)

## Acknowledgments

We are grateful to Dr. Piriyaporn Chongtrakool for providing a steers replicator to perform the agar dilution testing. We also thank the microbiology laboratory staff, Department of Microbiology at the Siriraj Hospital, Bangkok, Thailand, for collecting and providing clinical isolates for this study. We are also indebted to Mr. David Park for English-language editing.

## Author Contributions

**Conceptualization:** Natakorn Nokchan, Thidathip Wongsurawat, Perapon Nitayanon, Chanwit Tribuddharat.

**Data curation:** Natakorn Nokchan, Thidathip Wongsurawat, Piroon Jenjaroenpun.

**Formal analysis:** Natakorn Nokchan, Thidathip Wongsurawat, Piroon Jenjaroenpun, Perapon Nitayanon.

**Funding acquisition:** Natakorn Nokchan.

**Investigation:** Natakorn Nokchan, Perapon Nitayanon, Chanwit Tribuddharat.

**Methodology:** Natakorn Nokchan, Thidathip Wongsurawat, Piroon Jenjaroenpun, Perapon Nitayanon, Chanwit Tribuddharat.

**Project administration:** Natakorn Nokchan, Chanwit Tribuddharat.

**Resources:** Chanwit Tribuddharat.

**Software:** Thidathip Wongsurawat, Piroon Jenjaroenpun.

**Supervision:** Chanwit Tribuddharat.

**Validation:** Natakorn Nokchan, Thidathip Wongsurawat, Piroon Jenjaroenpun, Chanwit Tribuddharat.

**Visualization:** Natakorn Nokchan, Thidathip Wongsurawat, Piroon Jenjaroenpun.

**Writing – original draft:** Natakorn Nokchan.

**Writing – review & editing:** Chanwit Tribuddharat.

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
