## [Decision Letter · Decision Letter 0]

24 May 2022

PONE-D-22-11066Whole-genome sequence analysis of high-level penicillin-resistant strains and antimicrobial susceptibility of Neisseria gonorrhoeae clinical isolates from ThailandPLOS ONE

Dear Dr. Chanwit Tribuddharat,

Thank you for submitting your manuscript to PLOS ONE. After careful consideration, we feel that it has merit but does not fully meet PLOS ONE’s publication criteria as it currently stands. Therefore, we invite you to submit a revised version of the manuscript that addresses the points raised during the review process.

ACADEMIC EDITOR:

Please submit your revised manuscript by 8th July 2022. If you will need more time than this to complete your revisions, please reply to this message or contact the journal office at plosone@plos.org. Please include the following items when submitting your revised manuscript:A rebuttal letter that responds to each point raised by the academic editor and reviewer(s). You should upload this letter as a separate file labeled 'Response to Reviewers'.A marked-up copy of your manuscript that highlights changes made to the original version. You should upload this as a separate file labeled 'Revised Manuscript with Track Changes'.An unmarked version of your revised paper without tracked changes. You should upload this as a separate file labeled 'Manuscript'.If applicable, we recommend that you deposit your laboratory protocols in protocols.io to enhance the reproducibility of your results. Protocols.io assigns your protocol its own identifier (DOI) so that it can be cited independently in the future. For instructions see: https://journals.plos.org/plosone/s/submission-guidelines#loc-laboratory-protocols. Additionally, PLOS ONE offers an option for publishing peer-reviewed Lab Protocol articles, which describe protocols hosted on protocols.io. Read more information on sharing protocols at https://plos.org/protocols?utm_medium=editorial-email&utm_source=authorletters&utm_campaign=protocols.

We look forward to receiving your revised manuscript.

Kind regards,

Ayesha Sabah Rahman, PhD

Academic Editor

PLOS ONE

Journal Requirements:

4. PLOS requires an ORCID iD for the corresponding author in Editorial Manager on papers submitted after December 6th, 2016. Please ensure that you have an ORCID iD and that it is validated in Editorial Manager. To do this, go to ‘Update my Information’ (in the upper left-hand corner of the main menu), and click on the Fetch/Validate link next to the ORCID field. This will take you to the ORCID site and allow you to create a new iD or authenticate a pre-existing iD in Editorial Manager. Please see the following video for instructions on linking an ORCID iD to your Editorial Manager account: https://www.youtube.com/watch?v=_xcclfuvtxQ.

Reviewers' comments:

Reviewer's Responses to Questions

**Comments to the Author**

1. Is the manuscript technically sound, and do the data support the conclusions?

Reviewer #1: Yes

Reviewer #2: Partly

2. Has the statistical analysis been performed appropriately and rigorously? 

Reviewer #1: N/A

Reviewer #2: No

3. Have the authors made all data underlying the findings in their manuscript fully available?

Reviewer #1: Yes

Reviewer #2: Yes

4. Is the manuscript presented in an intelligible fashion and written in standard English?

Reviewer #1: Yes

Reviewer #2: Yes

5. Review Comments to the Author

Reviewer #1: This manuscript is well written and seem scientifically sound to me. I have listed below a number of comments and suggestions that I think should be addressed before publication in PLoS One.

Major Issues

Line 338. Sequence analysis of gonococcal plasmids - can these plasmids be assigned to an compatibility group (I think they are normally IncP-1) and / or a relaxase group e.g. using MOBscan?

Line 365. Figure 1 is so low resolution as to be unreadable - I am not sure if this is due to compression during production of the proof pdf, or because the authors supplied a low resolution image.

Lines 402-403. "The possible explanations for this result..." - I assume his has been double or triple checked? If not then it should be; if it has been then a measurement error seems unlikely?

Lines 540-541. "The plasmid backbones of conjugative plasmids could not be discriminated..." this should be reported in the Results section - but I'm not clear what this means? Surely with the whole plasmid genome sequence available a virtual restriction digestion could be carried out, or if sequences are available for prototypic Dutch / American plasmids then some sequence comparisons could be made to allow assignment?

Minor Issues

Lines 32-33. "Nanopore and Illumina sequencing characterized beta-lactam resistance determinants and plasmids..." please clarify this was WGS, not only plasmids and resistance genes.

Lines 43-44. "...carried a conjugative plasmid or the conjugative plasmid with the Dutch type tetM determinant" is confusing - please rewrite for clarity.

Line 48. "one strain"?

Line 76. "The penicillin resistance..." - delete "The".

Lines 78-79. "horizontal gene transfer of a beta lactamase plasmid" i.e. there is more than 1 possible plasmid ?

Line 80. "The chromosomal mutations..."

Line 88. "The beta-lactamase plasmid contains" - is there only a single "clone" / type known of this plasmid? If not, this should be plural "plasmids"?

Line 103. "carries up to 3 types of plasmid"?

Line 297. "However, an A (adenine) deletion was identified..." a figure or some text showing this 13 bp repeat sequence with mutation might be useful.

Line 331. "a QAATPAKQ motif"?

Line 346. Specify "The conjugative plasmids".

Line 380. "encoding a hypothetical protein"

Line 391. "resistance determinants and plasmids of..."

Line 407. Delete 2nd "was" - "Agar dilution was preferred because...". Also - it is unclear to me whether this is a CLSI recommendation?

Line 494-495. "The penicillin acylation rate was decreased to...". Please clarify - this was not shown here? "was previously found to be decreased..."?

Line 535. This sentence is unclear and should be re-written - you have not provided the first genomic analysis of all plasmids in N. gonorrhoeae strains from Thailand?

Reviewer #2: the manuscript titled; Whole-genome sequence analysis of high-level penicillin-resistant strains and

2 antimicrobial susceptibility of Neisseria gonorrhoeae clinical isolates from Thailand is a sound piece of scientific work with valuable input,

Please consider the comment given in the attached document to further improve the manuscript.

6. PLOS authors have the option to publish the peer review history of their article (what does this mean?). If published, this will include your full peer review and any attached files.

Reviewer #1: No

Reviewer #2: No

---

## [Author Response · Author response to Decision Letter 0]

19 Jun 2022

Journal Requirements

Answer: The manuscript has been revised according to the PLOS ONE style templates to meet PLOS ONE's style requirements.

Answer: Additional details regarding informed consent and statement of anonymized patient data have been added in ethical consideration section.

Answer: The minimal data set underlying the results has been added in our manuscript (provided as accession numbers) and submitted as Supporting Information files. This information has been added and described within our revised cover letter. For our Data Availability statement, all relevant data are within the paper and its Supporting Information files.

4. PLOS requires an ORCID iD for the corresponding author in Editorial Manager on papers submitted after December 6th, 2016. Please ensure that you have an ORCID iD and that it is validated in Editorial Manager. To do this, go to ‘Update my Information’ (in the upper left-hand corner of the main menu), and click on the Fetch/Validate link next to the ORCID field. This will take you to the ORCID site and allow you to create a new iD or authenticate a pre-existing iD in Editorial Manager. Please see the following video for instructions on linking an ORCID iD to your Editorial Manager account: https://www.youtube.com/watch?v=_xcclfuvtxQ.

Answer: an ORCID iD of the corresponding author has been updated.

Answer: Our reference list has been reviewed and updated. Changes to the reference list has been shown in the rebuttal letter.

Reviewer #1

Major Issues

1. Line 338. Sequence analysis of gonococcal plasmids - can these plasmids be assigned to an compatibility group (I think they are normally IncP-1) and / or a relaxase group e.g. using MOBscan?

Answer: Some of gonococcal plasmids can be assigned to incompatibility groups based on phylogenetic analysis of plasmid backbone, including origin of replication, replicase and partitioning genes. Though incompatibility tests should be carried out to give a definitive answer. As a preliminary result, phylogenetic analysis of the backbone sequences (TraI, TraG, TrbE and TrfA proteins) of the gonococcal conjugative plasmids showed that they belong to a novel IncP1 subfamily. In spite of reviewing the literature and in silico detection based on PCR-based replicon typing, incompatibility groups of gonococcal beta-lactamase (Rio/Toronto type) and cryptic plasmids could not be identified. It is noteworthy that the gonococcal Asia-type plasmid belongs to incompatibility group W and carries a silent IncFII determinant, in contrast to the Africa-type plasmid, belongs to the IncFII group. Nonetheless, there are no established plasmid incompatibility groups among the gonococcal plasmids. As for plasmid classification using relaxases, phylogenetic trees are constructed from the family alignments to allow the classification of conjugative transfer systems in six MOB families: MOBF, MOBH, MOBQ, MOBC, MOBP and MOBV. For this respect, functional analysis and annotation of all gonococcal plasmids are required to identify the relaxase genes. This notion is supported by MOBscan result as it cannot classify any of our gonococcal plasmids into any of the nine MOB families.

2. Line 365. Figure 1 is so low resolution as to be unreadable - I am not sure if this is due to compression during production of the proof pdf, or because the authors supplied a low resolution image.

Answer: Figure 1 was validated with the Preflight Analysis and Conversion Engine (PACE) digital diagnostic tool to ensure that it meets PLOS requirements. Therefore, the low-resolution issue may occur during the compression process of the file for reviewers.

3. Lines 402-403. "The possible explanations for this result..." - I assume his has been double or triple checked? If not then it should be; if it has been then a measurement error seems unlikely?

Answer: Since the agar dilution method has been done in duplicate to ensure the measured MICs, an MIC measurement error should be minimized, but not totally excluded. In addition, the different MICs among these isolates could be from different levels of blaTEM gene expression or a truncated 24 kDA TEM-1 enzyme. The sentence has been edited.

4. Lines 540-541. "The plasmid backbones of conjugative plasmids could not be discriminated..." this should be reported in the Results section - but I'm not clear what this means? Surely with the whole plasmid genome sequence available a virtual restriction digestion could be carried out, or if sequences are available for prototypic Dutch / American plasmids then some sequence comparisons could be made to allow assignment?

Answer: The sentence has been removed. We performed in silico BglI digestion with the gonococcal conjugative plasmids containing Dutch type tetM determinants and found that they showed a similar digestion pattern to the previously reported Dutch type plasmid. Consequently, using in silico BglI digestion the plasmid backbones of conjugative plasmids could be discriminated into American or Dutch type. All relevant details in the Materials and methods, Results, and Discussion were added and updated.

Minor Issues

5. Lines 32-33. "Nanopore and Illumina sequencing characterized beta-lactam resistance determinants and plasmids..." please clarify this was WGS, not only plasmids and resistance genes.

Answer: The sentence has been edited.

6. Lines 43-44. "...carried a conjugative plasmid or the conjugative plasmid with the Dutch type tetM determinant" is confusing - please rewrite for clarity.

Answer: The sentence has been rewritten to "All isolates with high-level penicillin resistance carried the conjugative plasmids with or without the Dutch type tetM determinant, the beta-lactamase plasmid (Rio/Toronto), and the cryptic plasmid."

7. Line 48. "one strain"?

Answer: It has been changed to “1 strain” following the American Medical Association Manual of Style.

8. Line 76. "The penicillin resistance..." - delete "The".

Answer: “The” has been deleted.

9. Lines 78-79. "horizontal gene transfer of a beta lactamase plasmid" i.e. there is more than 1 possible plasmid ?

Answer: It has been edited to “horizontal gene transfer of the beta-lactamase plasmids”.

10. Line 80. "The chromosomal mutations..."

Answer: It has been added.

11. Line 88. "The beta-lactamase plasmid contains" - is there only a single "clone" / type known of this plasmid? If not, this should be plural "plasmids"?

Answer: It is not a single plasmid clone. The sentence has been corrected.

12. Line 103. "carries up to 3 types of plasmid"?

Answer: It has been edited.

13. Line 297. "However, an A (adenine) deletion was identified..." a figure or some text showing this 13 bp repeat sequence with mutation might be useful.

Answer: The 13-bp inverted repeat sequence with an adenine deletion position in the mtrR promoter region has been added.

14. Line 331. "a QAATPAKQ motif"?

Answer: It has been edited to “a QAATPAKQ insertion” because this is an insertion mutation of QAATPAKQ motif.

15. Line 346. Specify "The conjugative plasmids".

Answer: It has been specified.

16. Line 380. "encoding a hypothetical protein"

Answer: It has been corrected.

17. Line 391. "resistance determinants and plasmids of..."

Answer: It has been corrected.

18. Line 407. Delete 2nd "was" - "Agar dilution was preferred because...". Also - it is unclear to me whether this is a CLSI recommendation?

Answer: It has been deleted. The disk diffusion and agar dilution methods were recommended by the CLSI for testing antimicrobial susceptibility in N. gonorrhoeae isolates. However, agar dilution susceptibility testing is the “gold standard” for susceptibility testing of N. gonorrhoeae. The suitable method depends on the time, availability of resources, and research aims.

19. Line 494-495. "The penicillin acylation rate was decreased to...". Please clarify - this was not shown here? "was previously found to be decreased..."?

Answer: This sentence referred to the previous report cited in reference (54).

20. Line 535. This sentence is unclear and should be re-written - you have not provided the first genomic analysis of all plasmids in N. gonorrhoeae strains from Thailand?

Answer: The given sentence has been rewritten by excluding overstatement.

Reviewer #2

Abstract/ Introduction

1. Adding statement to justify the importance of study regionally and globally;

As this study is from Thailand (the study region), giving a general impact of gonococcal infection and the effect of AMR in the study region would elaborate on the relevance and importance of the study locally and globally.

Answer: The statement regarding the importance of study regionally and globally has been added in the introduction section. Continued antimicrobial surveillance (including molecular studies) as in this study is necessary to detect patterns of resistance to ensure treatment efficacy against gonococcal infection at regional and national level. With regards to this surveillance, it can help to provide additional details on the emergence and dissemination of resistant gonococcal strains, so that international countries can make control and prevent these strains. This is due to the fact that most of ceftriaxone-resistant isolates have emerged in Asia and subsequently spread globally. Additionally, one of the gonococcal strains with ceftriaxone resistance combined with high-level resistance to azithromycin was linked to Thailand. Quality-assured local antimicrobial resistance surveillance data could also partially help to improve the WHO global gonorrhoeae treatment guidelines.

2. Justification of sample size/patient recruitment

Answer: The sample size was consulted with a statistician of our institution and calculated based on the prevalence of cefixime- or ceftriaxone-resistant N. gonorrhoeae isolates using a sample size application (called “n4Studies”). The first aim of this study was to investigate the presence of those isolates using antimicrobial susceptibility testing. Since cefixime- or ceftriaxone-resistant N. gonorrhoeae isolates have not been reported in Thailand and barely found in other countries, a statistician suggested that the more sample you collect, the more chance you will get these isolates. Therefore, 117 gonococcal isolates we had in the retrospective collection were used for the study. There is no patient recruitment.

3. Justification of only 03 strains analysis

Answer: To uncover penicillin resistance determinants and plasmid composition, we have selected 3 N. gonorrhoeae strains from a total of 18 strains based on their MIC to penicillin (MIC ≥ 128 μg/ml) and different NG-MAST STs. The number of gonococcal strains is limited due to time constraints and shrinking budgets. Furthermore, our purpose is to provide a brief overview and discussion of the results.

4. Abstract line 32; mtrR were performed. Need to define mtrR

Answer: Additional information has been added.

5. Line 38: notifying the significance of the genes/genetic characteristics will highlight the importance of the study further

Answer: The significance of genes and their mutations associated with antimicrobial resistance in N. gonorrhoeae has been highlighted in the introduction as well as the discussion section. 

6. 46, 47 The gonococcal population in Thailand showed high susceptibility to all antimicrobials 47 recommended for gonorrhoea treatment.

But the line 37: All isolates were susceptible to spectinomycin, can you clarify the comment in lines 46,47

Answer: The sentence in lines 46 and 47 is ambiguous, indeed it actually means that the gonococcal population in Thailand showed high susceptibility to antimicrobials currently recommended for gonorrhea treatment, ceftriaxone and azithromycin. However, this sentence has been edited.

7. 48: expanded and enhanced surveillance of antimicrobial susceptibility 49 and study of genetic resistance determinants are essential to improve treatment guidelines.

Can you clarify why this is needed by considering the actual prevalence of gonococcal infection prevalence in Thailand?

Answer: Gonorrhea is one of the most commonly reported STD in Thailand. Prevalence of gonorrhea was estimated to be 13.1 per 100,000 population in 2015 (as available). In general, gonococcal infections acquired in or from Asia represent most verified ceftriaxone treatment failures, and several ceftriaxone-resistant strains have emerged in Asia and subsequently spread globally. In Thailand, 0.5% of isolates showed resistance or decreased susceptibility to ceftriaxone. Taking these aspects into account, expanded and enhanced surveillance of antimicrobial susceptibility in N. gonorrhoeae is imperative to inform the Thailand national treatment guideline. Nonetheless, limited geographical representativeness of the isolates (mostly from Bangkok) and the lack of periodic antimicrobial susceptibility surveillance are a hassle.

Introduction

8. Line 74: What is the significance of gonococcal infection prevalence in Thailand?

Answer: The significance is in the lines 90-93 about emerging ESBL NG producers.

9. Line 90; (M182T) should be referred to as strain?

Answer: M182T is referred to an amino acid substitution. This information has been added.

10. Line 122: should bring to earlier and also compare with global statistics to show why this study is important to Thailand

Answer: This paragraph has been moved to earlier. The global cases of ceftriaxone-resistant gonococcal strains in other countries are reported and referred in the lines 65-71, comparing to Thailand where these strains have never been reported. Since ceftriaxone-resistant strains remain rare, a statement underlining the presence of these strains should be better used to elaborate. The importance of this study to Thailand is given in the lines 145-148. Additional details have been added in lines 67 and 125-126.

11. Line 155: do the participants considered have received the antibiotic treatment before the study? 

Answer The study was retrospective and no patient involvement. The clinical data on previous antibiotic exposure of patients are difficult to retrieve and largely unavailable; therefore, it has not been considered for this study.

12. What is the statistical method used for Patient data and Prevalence and penicillin susceptibility of PPNG isolate

Answer: No statistical methods were used for analyzing patient data and prevalence and penicillin susceptibility of PPNG isolates.

13. Line 255: Do the study participants give their consent if so mention this in this section.

Answer: No informed consent was required due to the retrospective nature of the study. In addition, this non-clinical study used bacterial isolates recovered from clinical samples taken as part of routine clinical laboratory examination and patient information was anonymized and deidentified prior to analysis. The ethical reviewing committee has approved the study as non-human research. This statement has been added in the ethical consideration section.

14. Line 181: interpretative criteria of the guidelines; can you clarify further with regards to MIC

Answer: The obtained MIC values were interpreted based on the interpretative criteria of the CLSI 2020 guidelines as follows: penicillin, S≤0.06, I=0.12-1, R≥2; tetracycline, S≤0.25, I=0.5-1, R≥2; ciprofloxacin, S≤0.06, I=0.12-0.5, R≥1; spectinomycin, S≤32, I=64, R≥128; cefixime, S≤0.25; ceftriaxone, S≤0.25. “CLSI” has been added to specify guidelines.

15. Line 183; What is the coverage of the N gonorrhoeae genome? Are the AMR determinants considered located in short or long read sequences? Justify the usage of both sequencing methodologies.

Answer: The coverage of each N. gonorrhoeae is the number of bases that cover the genome. It describes how frequently, on average, based from read coverage of the genome sequence (Table x1). The AMR genes of N. gonorrhoeae is covered by both short- (illumina) and long-reads (Nanopore) as shown in Table x2 and Figure x1. As example, we plotted read coverage of each AMR genes of N. gonorrhoeae strain CT530 in Figure x1 using IGV software. The methodology of de novo assembly was written in the method section. The AMR genes were based on the literature review and predicted from each assemble genome using Resistance Gene Identifier software (v.5.1.0, https://github.com/arpcard/rgi) against the Comprehensive Antibiotic Resistance Database (v.3.0.9, https://card.mcmaster.ca/). Whole-genome sequencing with both Illumina and Nanopore platforms was performed to generate complete genome sequences of N. gonorrhoeae. As a result, we can obtain the full-length of all antimicrobial resistance genes and the entire plasmid sequences for analysis. 

Table x1. The read coverage of the N. gonorrhoeae genome

STRAIN GENOMESIZE ILLUMINA_TOTAL_BASES ILLUMINA_COV NANOPORE_TOTAL_BASES NANOPORE_COV

CT530 2220590 1546958433 697 488748095 220

CT532 2216438 1469088082 663 472676167 213

CT602 2211092 1747893851 791 1118067306 506

Table x2. The read coverage of AMR genes

CONTIG GENE CT530 CT532 CT602

 Nanopore Illumina Nanopore Illumina Nanopore Illumina

PLASMID blaTEM-1-WT 24826 16869 33518 24732 52891 30099

CHROMOSOME dacB (PBP3)-WT 80 624 72 570 246 653

CHROMOSOME dacC-WT 89 562 58 479 248 647

CHROMOSOME mtrR-WT 83 537 73 480 222 546

CHROMOSOME pbpG (PBP4)-WT 84 565 59 498 244 600

CHROMOSOME penA (PBP2)-WT 89 632 65 560 216 658

CHROMOSOME pilQ-WT 93 589 64 536 212 647

CHROMOSOME ponA (PBP1)-WT 101 620 65 580 237 681

CHROMOSOME porB1b-WT 113 686 75 654 273 755

Figure x1. The example of AMR genes in N. gonorrhoeae strain CT530

16. Line 465 what is the significance of the MLST ST8143, clarify

Answer: In Taiwan, the MLST ST8143 isolates were abruptly emerged after 2012. These isolates may follow the pattern of ST1901 and become highly prevalent locally and globally; therefore, they should be monitored. Of note, the circulating ST1901 strains generally exhibited higher MICs for both ceftriaxone and azithromycin. Moreover, the MLST ST8143 isolates were predominantly identified in the United States and less found in Ireland. The significance of the MLST ST8143 has been added.

17. Line 528: is there a positive correlation between the MIC and the corresponding genetic determinants/ it is not very clear

Answer: There is no experiment to show a positive correlation between the MIC and the penicillin resistance determinants in this study. Nonetheless, we made an attempt to correlate the high penicillin MIC value (≥ 128 μg/ml) of 3 representative gonococcal strains with well-known penicillin resistance determinants (penA, mtrR, penB and blaTEM-135). All specified mutations in these resistance determinants stated in the sentence have been previously studied and shown to increase penicillin MICs.

18. Line 535: how to justify the statement; We provide the first genomic analysis of all plasmids in N. gonorrhoeae strains from Thailand. Clarify it further

Answer: This statement is overstated in terms of novelty. The sentence has been edited.

19. Line 555: it is worth mention the previous antibiotic treatment data is also not available

Is the risk factors referring to the patients?

Answer: The lack of previous antibiotic treatment data of patients has been added. The risk factors refer to the patients. Additional details have been added.

---

## [Decision Letter · Decision Letter 1]

6 Jul 2022

Whole-genome sequence analysis of high-level penicillin-resistant strains and antimicrobial susceptibility of Neisseria gonorrhoeae clinical isolates from Thailand

PONE-D-22-11066R1

Dear Dr. Chanwit Tribuddharat,

We’re pleased to inform you that your manuscript has been judged scientifically suitable for publication and will be formally accepted for publication once it meets all outstanding technical requirements.

Kind regards,

Ayesha Sabah Rahman, PhD

Academic Editor

PLOS ONE

Reviewers' comments:

Reviewer's Responses to Questions

**Comments to the Author**

1. If the authors have adequately addressed your comments raised in a previous round of review and you feel that this manuscript is now acceptable for publication, you may indicate that here to bypass the “Comments to the Author” section, enter your conflict of interest statement in the “Confidential to Editor” section, and submit your "Accept" recommendation.

Reviewer #1: All comments have been addressed

Reviewer #2: All comments have been addressed

2. Is the manuscript technically sound, and do the data support the conclusions?

Reviewer #1: Yes

Reviewer #2: Yes

3. Has the statistical analysis been performed appropriately and rigorously? 

Reviewer #1: I Don't Know

Reviewer #2: Yes

4. Have the authors made all data underlying the findings in their manuscript fully available?

Reviewer #1: Yes

Reviewer #2: Yes

5. Is the manuscript presented in an intelligible fashion and written in standard English?

Reviewer #1: Yes

Reviewer #2: Yes

6. Review Comments to the Author

Reviewer #1: The authors have thoroughly and appropriately addressed all my points from the previous round of review.

Reviewer #2: The authors have clearly answered the comments raised by the reviewer. revised comments were appropriately added to the revised manuscript.

7. PLOS authors have the option to publish the peer review history of their article (what does this mean?). If published, this will include your full peer review and any attached files.

Reviewer #1: No

Reviewer #2: **Yes: **Dr Shivanthi Samarasinghe
